# Management of Traumatic and Non-Traumatic Cerebrospinal Fluid Rhinorrhea—Experience from Three Southeast Asian Countries

**DOI:** 10.3390/ijerph192113847

**Published:** 2022-10-25

**Authors:** Farah Dayana Zahedi, Somasundaram Subramaniam, Pornthep Kasemsiri, Chenthilnathan Periasamy, Baharudin Abdullah

**Affiliations:** 1Department of Otorhinolaryngology-Head & Neck Surgery, Faculty of Medicine, Universiti Kebangsaan Malaysia, Kuala Lumpur 56000, Malaysia; 2Department of Otolaryngology–Head and Neck Surgery, National University of Singapore, Singapore 119077, Singapore; 3Department of Otolaryngology–Head and Neck Surgery, Ng Teng Fong General Hospital, Singapore 609606, Singapore; 4Department of Otolaryngology–Head and Neck Surgery, Srinagarind Hospital, Faculty of Medicine, Khon Kaen University, Khon Kaen 40000, Thailand; 5Department of Otorhinolaryngology–Head and Neck Surgery, Penang General Hospital, George Town 10990, Malaysia; 6Department of Otorhinolaryngology–Head and Neck Surgery, School of Medical Sciences, Universiti Sains Malaysia, Kubang Kerian 16150, Malaysia

**Keywords:** cerebrospinal fluid rhinorrhea, cerebrospinal fluid leak, rhinorrhea, skull base, cribriform plate, endoscopic surgical procedure

## Abstract

Background: Cerebrospinal fluid (CSF) rhinorrhea requires proper management to avoid disastrous consequences. The objectives of this study were to ascertain the patient characteristics, etiologies, sites of defect, skull base configurations, methods of investigation, and management outcomes of CSF rhinorrhea. Methods: A retrospective study was performed over 4 years involving three surgeons from Malaysia, Singapore, and Thailand. Hospital records were reviewed to determine the patients’ characteristics, the causes and sites of leaks, methods of investigation, skull base configurations, choices of treatment, and outcomes. Results: A total of 15 cases (7 traumatic and 8 non-traumatic) were included. Imaging was performed in all cases. The most common site of leakage was the cribriform plate (9/15 cases). The mean ± SD of the Keros heights were 4.43 ± 1.66 (right) and 4.21 ± 1.76 mm (left). Type II Keros was the most common (60%). The mean ± SD angles of the cribriform plate slope were 51.91 ± 13.43 degrees (right) and 63.54 ± 12.64 degrees (left). A class II Gera configuration was the most common (80%). All except two patients were treated with endonasal endoscopic surgical repair, with a success rate of 92.3%. A multilayered repair technique was used in all patients except one. The mean ± SD postoperative hospital stay was 9.07 ± 6.17 days. Conclusions: Non-traumatic CSF rhinorrhea outnumbered traumatic CSF rhinorrhea, with the most common site of leak at the cribriform plate. Imaging plays an important role in investigation, and Gera classification appears to be better than Keros classification for evaluating risk. Both conservative and surgical repairs are practiced with successful outcomes. Endonasal endoscopic CSF leak repair is the mainstay treatment.

## 1. Introduction

The treatment for cerebrospinal fluid (CSF) rhinorrhea is formidable and warrants a collaborative effort from an interdisciplinary team with expert care. Disastrous consequences from ascending infection may ensue when the condition is left untreated. CSF rhinorrhea is conventionally classified into traumatic and non-traumatic causes [1]. Under the umbrella of the traumatic etiology, CSF rhinorrhea can be subclassified into iatrogenic or non-iatrogenic, while non-traumatic causes are secondary to tumors, hydrocephalus, and meningitis. The recent systematic review by Xie et al. showed that the prevalence of traumatic CSF rhinorrhea was 44%, the prevalence of iatrogenic CSF rhinorrhea was 12%, and the prevalence of spontaneous CSF rhinorrhea was 28% [2]. In the absence of an identifiable etiology, CSF rhinorrhea is categorized as a primary spontaneous leak and is often associated with raised intracranial pressure and idiopathic intracranial hypertension (IIH) [3]. The risk factors of spontaneous CSF rhinorrhea are obesity, female gender, and obstructive sleep apnea [4]. Previous studies showed that patients with spontaneous CSF rhinorrhea have an elevated body mass index, ranging approximately from 35 to 38 kg/m^2^ [5,6]. Spontaneous CSF rhinorrhea predominantly occurs in middle-aged women, with a typical age of presentation around 45–65 years [6,7].

The definitive management of CSF rhinorrhea is primarily surgery, utilizing mostly endoscopic endonasal repair techniques, with open approaches practiced in a small number of cases [8]. Conservative management consists of bed rest and head elevation, and acetazolamide may be beneficial in some instances, particularly in non-iatrogenic traumatic leaks. The employment of non-surgical treatment with active surveillance in traumatic CSF rhinorrhea is based on the rationale that granulating tissue around the site of the defect may seal off the leak [9]. The identification of the site and size of defects in CSF rhinorrhea is critical in determining the management plan. To date, there are two well-known classification systems of the anterior skull base, namely the Keros and Gera classifications [10,11]. The association of these classifications with traumatic CSF rhinorrhea was well-described [12,13,14].

The early and accurate diagnosis of a CSF leak can lead to better treatment outcomes. It can also help prevent complications. Although much has been described about the management of CSF leaks, there was a lack of information on the presentation and management of CSF rhinorrhea in Southeast Asian countries. To address this gap, the objectives of this study were to ascertain the patient characteristics, etiologies, sites of defect, skull base configurations, methods of investigation, and management outcomes of CSF rhinorrhea from three different countries in Southeast Asia, namely Malaysia, Singapore, and Thailand. The three selected countries have similar population demography, socioeconomic status, and health care systems and could reliably reflect the management practice for CSF leaks in Southeast Asian countries in general. The information obtained would be helpful to understand and improve the management of patients with CSF rhinorrhea.

## 2. Materials and Methods

A retrospective study of patients with CSF rhinorrhea over 4 years from three centers in Southeast Asia was performed. In view of the retrospective case series and multicenter experience of this study, ethical approval exemption was granted by the institutions. All patients with CSF rhinorrhea and being treated at three tertiary centers in Malaysia, Thailand, and Singapore within the study period were included. The involved study groups were three surgeons from Penang General Hospital (Malaysia), National University Health System (Singapore), and Srinagarind Hospital (Thailand). Patients that had CSF leaks due to congenital anomalies, incomplete records, and untraceable investigations and those who were lost to follow-up were excluded.

Following the review of the three hospitals’ records, a total of 15 patients fulfilled the selection criteria. Nine were males, and six were females. Of these, seven cases were traumatic (three iatrogenic and four non-iatrogenic), and eight cases were non-traumatic (where four of the eight were suspected to be due to IIH). From all the cases of spontaneous CSF rhinorrhea, 62.5% were female, and 80% of the females with spontaneous CSF rhinorrhea were attributed to IIH. The age, site of leak, anterior skull base configuration, treatment (including surgical repair technique), and outcomes were recorded. Data, including the duration of the hospital stay following surgery and the duration of follow-up, were documented.

The anterior skull base was evaluated according to the Keros and Gera classifications [10,11]. Multiplanar computed tomographic images of the paranasal sinuses were used. Bone view images in coronal sections were selected for measurement. The right and left sides were measured separately. The Keros classification measures the depth of the olfactory fossa (vertical height of the lateral lamella of the cribriform plate). It has three types: type I (depth of 1–3 mm), type II (depth of 4–7 mm), and type III (depth of 8–16 mm). The Gera classification measures the angle formed by the lateral lamella of the cribriform plate and the continuation of the horizontal plane passing through the cribriform plate. It includes three classes: class I (>80 degrees, low risk), class II (45–80 degrees, medium risk), and class III (<45 degrees, high risk). Descriptive statistical methods were used to analyze the data. Fisher’s exact test was used to determine if there was a significant association between the categorical variables. *p* < 0.05 was considered significant.

## 3. Results

### 3.1. Patient Characteristics

The characteristics of the patients that presented with CSF rhinorrhea are shown in Table 1.

### 3.2. Methods in the Confirmation of Site of Leak

Apart from the clinical assessment, the confirmation of the site of leak was mainly made by imaging. Both high-resolution CT and MRI of the paranasal sinuses and skull base were used to localize the site of leak.

The use of imaging identified that the most common site of the leak was the cribriform plate (9 of 15 cases), followed by the sphenoid and frontal sinus posterior table (3 cases each). From the nine patients with identifiable defects in cribriform plate, more than half had spontaneous CSF rhinorrhea, while the rest had traumatic CSF rhinorrhea (two had iatrogenic CSF rhinorrhea and another two had non-iatrogenic CSF rhinorrhea).

Intraoperative intrathecal fluorescein was employed in four cases to aid in the intraoperative localization of the site of the leak, either in cases where the site of the CSF leak was not identified pre-operatively or when there was a possibility of multiple sites, especially in the non-iatrogenic traumatic cases. Beta-2 transferrin was used in two cases that presented with intermittent leaks, where the diagnosis was in doubt.

### 3.3. The Anterior Skull Base Configuration

The CT scans of the 15 cases (with 30 sides) were analyzed for the height difference between the olfactory groove and the ethmoidal roof (Keros classification) (Table 2). The mean ± SD of the Keros heights were 4.43 ± 1.66 mm on the right side (range 1.3–7.0 mm) and 4.21 ± 1.76 mm on the left side (range 1.2–6.8 mm) (Figure 1). When comparing the sides within each case, there was a mean ± SD difference in height between the left and right side of 0.81 ± 0.68 mm, with a range of 0.1–2.5 mm.

From all studied cases, 30 sides were taken for the analysis. In total, 12 sides (40%) were type 1 Keros configurations, and 18 sides (60%) were type II Keros configurations. None of the sides had a type III Keros configuration (Table 2). An analysis of the slope of the skull base at the cribriform plate (Gera classification) noted mean ± SD angles of 51.91 ± 13.43 degrees on the right side (range 23.0 to 72.5 degrees) and 63.54 ± 12.64 degrees on the left side (range 35.0 to 78.7 degrees). When comparing the sides within each case, there was a mean difference of 11.90 ± 9.38 degrees between the sides, with a range difference of 1.6–33.0 degrees. This large mean difference between the sides suggests that the cases in this series had significant asymmetry between the two sides of the skull base (Table 2). Most of the sides in this study had class II Gera configurations (80%), followed by class III Gera configurations (20%). None of the sides had a class I Gera configuration (Table 3).

We subdivided the cases into traumatic and non-traumatic CSF rhinorrhea for comparison (Table 4). The most common site of leak in both groups was the cribriform plate. However, no significant difference in the site of leak was seen between the groups (*p* = 0.64). While no cases in either group were Keros III, there was a significant difference in the Keros I and II classification between the traumatic and non-traumatic cases (*p* = 0.02). Type I Keros predominated in the traumatic cases, and type II Keros predominated in the non-traumatic cases. No significant difference was noted between the groups for the Gera classification (*p* = 1.00), with both having Gera II as the most common type.

### 3.4. Treatment and Outcomes

Intraoperatively, intrathecal fluorescein was used in 3 out of 13 patients to aid with the identification of the site of the CSF leak. Endonasal endoscopic surgical repair of the CSF rhinorrhea was performed in all patients except two (who were treated conservatively). The two patients that were treated conservatively had spontaneous CSF rhinorrhea (one had idiopathic CSF rhinorrhea, and one had IIH). None of the patients had an open surgical repair. For all performed endonasal endoscopic surgical repairs of CSF rhinorrhea, all three surgeons shared the same technique of using multilayered repair, except in one patient (monolayered repair due to surgical difficulties). A graft was preferred for repairing the skull base defect (middle turbinate graft: 6/13, fat graft: 5/13, and nasal floor graft: 1/3). From the five patients that used a nasoseptal flap, four were in combination with the fat plug technique. Surgical glue was used in 9 of 13 patients (69.2%). One of the surgeons (Singapore) did not use surgical glue in four patients, with no postoperative CSF rhinorrhea reported (Figure 2). A lumbar drain was used in 2 of 15 patients.

All but one patient had a successful surgical repair (92.3%). In one patient, a delayed leak presented a month after the first endonasal endoscopic CSF leak repair using a multilayered closure combined with a nasoseptal flap, a fat graft, and tissue glue. The second repair was performed using a similar technique with a successful outcome. There was no recurrent CSF rhinorrhea in two patients that were treated conservatively. The mean ± SD postoperative hospital stay was 9.07 ± 6.17 days (range 3–24 days). The mean ± SD follow-up duration following surgery was 5.15 ± 4.77 months (range 1–15 months). There was no significant difference in recurrent leaks between conservative management and endoscopic multilayered closure (*p* = 1.00) (Table 5).

## 4. Discussion

Due to the serious potential complications of CSF rhinorrhea, it must be appropriately recognized, and the site of leak must be accurately identified for prompt management to be instituted. The leaks occur due to an abnormal connection between the subarachnoid space and a defect at skull base that leads to communication between the intracranial and sinonasal areas. The causes of CSF rhinorrhea are classified as traumatic and non-traumatic. Both types were found in our patients. Traumatic causes were recognized to be more common than the non-traumatic causes [2,15]. Conversely, we observed that non-traumatic CSF leaks (53.3%) represented the most common cause of CSF rhinorrhea in the present case series. This could be attributed to the study design, where patients with incomplete records or untraceable imaging were not included.

In a study involving 40 patients with CSF rhinorrhea, Sannareddy et al. observed the following characteristics. The prevalence of spontaneous CSF leaks was more common in middle-aged females, whereas post-traumatic CSF leaks were common in young adult males [16]. Typically, the majority of the patients that presented with spontaneous CSF leaks in our series were middle-aged females with IIH. IIH is a known cause of primary spontaneous CSF rhinorrhea [17,18,19,20]. In our series, there was a high success rate with endoscopic repair, which is very much comparable with studies conducted in Western countries [1].

Interestingly, iatrogenic trauma was more common (57%) in the traumatic group than the non-iatrogenic cause (43%). As this prevalence varies from study to study, an agreement cannot be reached on the comparison between iatrogenic and non-iatrogenic trauma [15,21]. Due to the increased number of endoscopic sinus and skull base surgeries being performed, a rising trend of iatrogenic CSF leaks is noted. A meta-analysis by Alkis et al. on the causes of CSF leaks in 1685 cases revealed iatrogenic trauma to be more common than non-iatrogenic trauma [22]. Our finding most likely reflects this current trend.

In the traumatic group, male gender (86%) predominated compared to female gender (14%). In Southeast Asian countries, men outnumber women as the bread winners of their families, and frequently traveling to work or working on the road exposes them to a higher risk of motor vehicle accidents. In patients experiencing iatrogenic leaks, the possibility of men having an anatomical configuration at increased risk of such an occurrence cannot be excluded. A detailed assessment of the skull base anatomy, including the anterior ethmoidal roof and cribriform plate, is a pivotal critical step in the pre-operative evaluation of every patient to avert such complications.

Imaging plays a key role in the accurate localization of skull base defects in patients with CSF rhinorrhea. A high-resolution CT of the brain and paranasal sinus is indispensable and represents the standard investigation to determine the site of the defect and for surgical mapping, while the additional use of MRI is valuable in spontaneous CSF rhinorrhea cases. High-resolution CT scan had sensitivity and specificity values of 92% and 100%, respectively, compared to MRI with sensitivity and specificity values of 87% and 100%, respectively, in pinpointing the skull base defect [23]. A recommended investigation in CSF rhinorrhea is a high-resolution CT scan as first-line imaging due to its sufficient diagnostic characteristics, low cost, and noninvasive nature, while MRI, which had better test characteristics but a greater cost, is reserved as a second-line investigation when needed [2]. All of our cases had CT scans as an imaging modality in detecting the skull base defect, and 10 cases had an additional MRI, mainly for spontaneous CSF rhinorrhea. Additional investigations using beta-2 transferrin and intrathecal fluorescein were also performed in selected patients in our series. Beta-2 transferrin is costly and unavailable in most Southeast Asian countries, which precludes its use as a standard routine test for CSF rhinorrhea [24]. Intrathecal fluorescein is an invasive diagnostic procedure that appears to be less useful without sufficient expertise and was shown to be inferior to imaging [2]. In our series, beta-2 transferrin and intrathecal fluorescein were employed in cases where the diagnosis was doubtful, identification of the leak on imaging failed, or when there was a possibility of the involvement of multiple sites. Impaired consciousness without a history of trauma is a common reason for emergency department (ED) visits. Among critically ill patients with a history and physical findings suggestive of a cerebrovascular accident (CVA), it may be difficult to differentiate between structural and non-structural causes for their condition. In critically ill patients with acutely altered levels of consciousness but without a history of trauma, a CSF-LDH value < or = 40 IU/L was associated with a non-structural pathology [25].

The Keros classification of the anterior skull base is acknowledged as an important assessment, especially as a pre-operative evaluation in endoscopic sinus and skull base surgeries, where surgeons can stratify cases as low- or high-risk. A high depth of olfactory fossa (type III Keros) is conventionally associated with a greater risk of an anterior skull base traumatic CSF leak [10]. Nonetheless, a radiological study found that Keros classification alone is not enough to identify the high-risk area at the skull base and the ethmoidal roof [26]. As the Keros classification does not take into account the sloping level of the ethmoidal roof relative to the cribriform plate, the Gera classification was introduced [11]. By measuring the angle formed by the lateral lamella of the cribriform plate and the continuation of the horizontal plane passing through the cribriform plate, the skull base was divided into three classes: class I (low risk), class II (medium risk), and class III (high risk). Notably, the Gera classification has higher sensitivity and specificity, (79.2% and 96%, respectively) compared to the Keros classification (4.2% and 84%, respectively) [12]. We found that none of the cases had type III Keros, and only 6 from 30 sides had a class III Gera classification in the overall series. Five of the six sides of the paranasal sinuses in the samples with iatrogenic trauma (three patients) had type I Keros. Five of seven patients had class II Gera bilaterally, one had class III Gera bilaterally, and one had class II and class III Gera on each side. The case with class III Gera bilaterally had type I Keros bilaterally. Though this case series had a limited number of patients, it is tempting to speculate that this finding may suggest that the Gera classification is better than the Keros classification for evaluating the risk of traumatic CSF leak.

A comprehensive review of the literature involving 1622 patients identified that the most common site of leak was the ethmoid roof (33%), followed by the cribriform plate (28%), sphenoid sinus (22%), and frontal sinus (8%) [2]. In the present series, the most common site of CSF leak was the cribriform plate (60%), followed by the sphenoid and the posterior table of frontal sinus (20% each). Even though the frequency of each site varies, the similarities of the leak sites involved are not a coincidence. The ethmoid roof and cribriform plate, which form the thinnest structure at the skull base, and the lateral recess of the sphenoid sinus, due to over pneumatization of the sphenoid laterally, are weak and vulnerable to developing CSF leaks. Any increase in pressure intracranially due to trauma or spontaneous conditions has a propensity to inflict damage at the weakest structures first, whereas an incautious surgical dissection at these regions could inadvertently lead to the same aftermath.

CSF leaks are not uncommon after a base of skull fracture. Currently, there is no standardized algorithm for the investigation and management of post-traumatic CSF leaks. The diagnosis of a base of skull fracture and any resultant CSF leak can be challenging. Therefore, a combination of biochemical and radiological studies is needed to optimize the diagnosis of this condition. Post-traumatic CSF leaks are generally treated conservatively, and a majority of them resolve without further surgical management. However, for patients who are refractory to such treatments, surgical closure of the CSF fistula is necessary. The surgical obliteration of CSF leaks can be challenging and requires the involvement of multiple surgical specialties such as neurosurgery, otolaryngology, and maxillofacial surgery [27].

Both conservative medical management and surgical repair were employed in this study with successful outcomes. The aim of conservative medical management is to decrease active flow through the leaking site, thereby reducing the CSF pressure and allowing the healing of the defect to seal the leak. These measures usually require a prolonged hospital stay to achieve the desired result. A review asserted that the overall termination of CSF leaks with conservative treatment is 39.5% at 3 days, which will improve up to 85% at 1 week or more [15]. Two cases of non-traumatic CSF rhinorrhea were treated by intravenous antibiotics, acetazolamide, and bed rest with stool softeners. These patients had a mean hospital stay of 20 days. It is noteworthy that the use of acetazolamide therapy as a primary treatment option for spontaneous CSF rhinorrhea has been shown to pre-empt surgery in 31.3% of patients [28].

The modern techniques of endonasal endoscopic surgery have improved the success rate of CSF repair tremendously to over 95% [29,30]. Surgical treatment was required in thirteen of the cases by the endonasal endoscopic surgical approach. One patient developed recurrent CSF rhinorrhea 1 month postoperation, which was successfully treated by a second endonasal endoscopic surgery. Although the open approach was counselled in all cases for repair, none was necessary. The patients that were treated surgically had a mean hospital stay of 9 days.

All three surgeons used a similar technique of multilayered repair, except for with one patient due to surgical difficulties. However, the usage of grafts or pedicled flaps varied according to surgeons’ preferences. A wide variety of repair materials are available for the endoscopic repair of CSF leaks. Autologous fat, collagen matrix, free mucosal grafts, and tissue sealant are among the popular options, and less commonly a pedicled mucosal flap is used for postoperative persistent or recurrent leaks [31,32,33,34]. Most centers worldwide treat CSF rhinorrhea with endonasal endoscopic surgery involving a multilayered repair technique, with an open approach reserved for complex cases or a concomitant intracranial pathology. CSF rhinorrhea, by and large, does not require lumbar drainage, except where there is an individual surgeon preference or in recurrent leaks [35,36].

A systematic review and meta-analysis on the outcomes of endoscopic CSF leak repair reported that the overall success rate of endonasal endoscopic CSF leak repair was 90% for primary repair and 97% for secondary repair, with a low complication rate of less than 0.03% [22]. Another meta-analysis involving 2000 cases of primary CSF leak repair reached the same conclusion, with a success rate of 90.1% [21]. These rates are superior compared to the success rates of 86% in craniotomy and extracranial methods for the surgical repair of CSF leaks [37]. A review of recent studies published over the past 5 years showed a high success rate with zero or very minimal immediate postoperative complications (Table 6) [8,30,31,32,38,39,40,41,42,43,44,45,46,47,48,49]. The variable success rates of endoscopic repair most likely reflect the surgeons’ technical experience, the timing of management, and the nature of the leak (high-pressure versus low-pressure). This signifies that endoscopic leak repair represents the standard of care for most centers managing CSF rhinorrhea.

### Limitations

The limitation of this case series was the small sample size that only represented the experience of three centers from three countries in Southeast Asia. Due to this and the retrospective nature of this study, the findings must be interpreted with caution. Further data of CSF leaks from other centers in Southeast Asian countries are required to corroborate the findings from this case series.

## 5. Conclusions

Non-traumatic cases outnumbered traumatic CSF rhinorrhea in this case series. Both types shared a common site of leak at the cribriform plate. Imaging using CT and MRI of the brain and paranasal sinuses played an important role in confirming the CSF leak. The assessment of the anterior skull base by the Gera classification appears to be better than the Keros classification for evaluating the risk of a traumatic CSF leak. Both conservative medical treatment and endonasal endoscopic surgical repair were practiced to treat CSF rhinorrhea with successful outcomes. Endonasal endoscopic CSF leak repair is the mainstay of treatment for CSF rhinorrhea using multilayer repair techniques.

## Figures and Tables

**Figure 1 ijerph-19-13847-f001:**
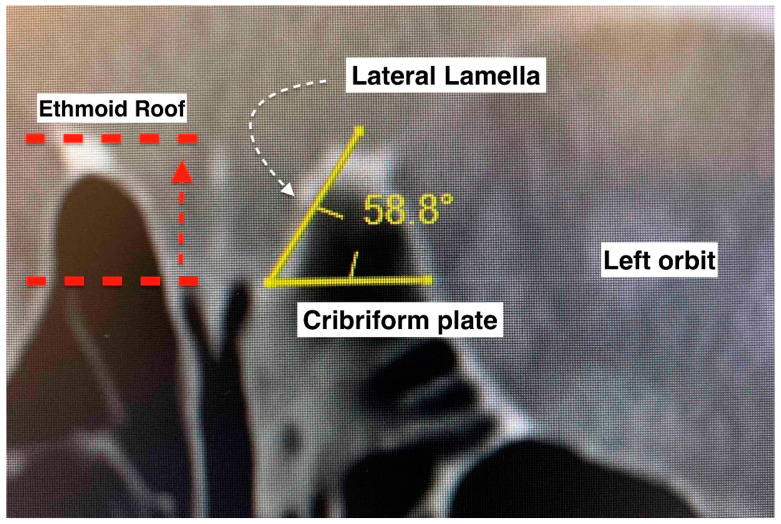
Keros and Gera classifications of the anterior skull base configuration. Keros compares the vertical height difference between the cribriform plate and the ethmoid roof. Gera is the calculation of the angulation between the horizontal plane of the cribriform plate floor and the lateral lamella. The smaller the angle, the higher the risk of inadvertent penetration.

**Figure 2 ijerph-19-13847-f002:**
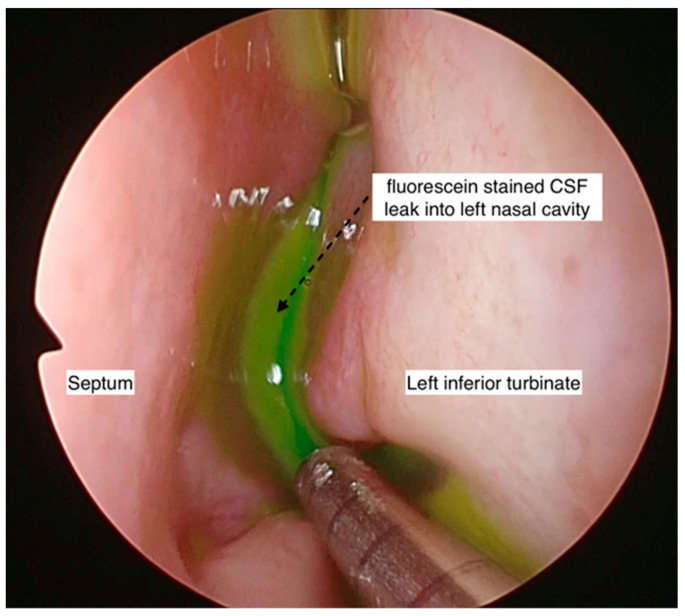
Endoscopic photo of left nasal cavity showing fluorescein-stained CSF within the nasal cavity. This aids in the identification of the exact site of the CSF leak or if there are multiple sites of leak.

**Table 1 ijerph-19-13847-t001:** Patient demographics (F: Female, M: Male, IIH: Idiopathic Intracranial Hypertension, Rt: Right, Lt: Left).

Case	Age	Gender	Diagnosis	Cause of CSF Leak	Site of Leak	Pre-Op Imaging
1	46	F	IIH	Spontaneous	Cribriform plate defect	CT and MRI
2	45	M	Postcraniotomy leak	Iatrogenic Trauma	Posterior table of frontal sinus	CT
3	33	M	Previous trauma with meningitis	Non-Iatrogenic Trauma	Cribriform plate defect	CT
4	29	M	Industrial trauma	Non-Iatrogenic Trauma	Suprasellar/tuberculum sellae fracture line	CT
5	40	F	IIH	Spontaneous	Anterior cranial fossa defect (cribriform plate)	CT and MRI
6	59	M	Idiopathic	Spontaneous	Lt frontal sinus defect	CT and MRI
7	66	F	Idiopathic	Spontaneous	Rt anterior cranial fossa floor defect (cribriform plate)	CT and MRI
8	61	F	IIH	Spontaneous	Rt sella defect at the anterior wall	CT and MRI
9	61	F	IIH	Spontaneous	Sphenoid sinus defect	CT and MRI
10	26	M	Idiopathic	Spontaneous	Cribriform plate defect	CT and MRI
11	81	M	Inverted papilloma	Iatrogenic Trauma	Rt posterior table frontal sinus	CT and MRI
12	77	M	Nasal polyp	Iatrogenic Trauma	Rt cribriform plate	CT
13	20	M	RTA	Non-Iatrogenic Trauma	Rt cribriform plate	CT and MRI
14	11	M	Frontoethmoid mucocele	Spontaneous	Rt cribriform plate	CT and MRI
15	16	F	Epidural hematoma/Craniotomy	Iatrogenic Trauma	Lt cribriform plate	CT

**Table 2 ijerph-19-13847-t002:** Keros vs. Gera classification. Keros I—height difference of <4 mm between cribriform plate (CP) and ethmoid roof (ER). Keros II and III—4–8 mm and >8 mm height differences, respectively. Gera: class I (>80 degrees, low risk), class II (45 to 80 degrees, medium risk), and class III (<45 degrees, high risk), according to the theoretical risk of iatrogenic injuries. Angulation is calculated between the intersection of the horizontal line of the floor of the CP and the lateral lamella of the CP.

Case	Keros Classification (mm)		L/R Difference (mm)	Gera Classification (Degrees)		L/R Difference (Degrees)
	R	L		R	L	
1	II (4.7)	II (4.1)	0.5	II (72.2)	II (78.7)	6.5
2	I (3.7)	II (4.3)	0.6	II (68.5)	II (73.5)	5.0
3	II (4.0)	I (3.2)	0.8	II (60.5)	II (68.4)	7.9
4	I (1.3)	I (1.2)	0.1	III (30.5)	III (35.0)	4.5
5	II (6.5)	II (4.8)	1.7	II (60.9)	II (76.7)	15.8
6	II (4.2)	I (1.7)	2.5	III (44.4)	III (42.8)	1.6
7	II (5.6)	II (5.8)	0.2	III (42.1)	II (66.8)	24.7
8	II (6.3)	II (6.2)	0.1	II (60.0)	II (73.1)	13.1
9	II (7.0)	II (6.2)	0.8	II (51.6)	II (63.0)	11.4
10	II (5.8)	II (6.8)	1.0	II (48.6)	II (76.1)	27.5
11	I (2.6)	I (3.0)	0.4	II (53)	II (59)	6.0
12	I (2.1)	I (2.8)	0.7	III (23)	II (56)	33.0
13	II (4.7)	I (3.7)	1	II (53)	II (61)	8.0
14	I (3.2)	I (3.0)	0.2	II (47)	II (54)	7.0
15	II (4.7)	II (6.3)	1.6	II (63)	II (69)	6.0

**Table 3 ijerph-19-13847-t003:** Comparison of sides of anterior skull base according to Keros and Gera classifications.

Type/Class	Keros *n* (%)	Gera *n* (%)
I	12 (40)	0 (0)
II	18 (60)	24 (80)
III	0 (0)	6 (20)
Total (sides)	30 (100)	30 (100)

**Table 4 ijerph-19-13847-t004:** Comparison of the site of leak and the Keros and Gera classifications in traumatic and non-traumatic CSF rhinorrhea.

	Traumatic	Non-Traumatic	*p* Value *
*Site of leak*	**Cases (*n* = 7), *n* (%)**	**Cases (*n* = 8), *n* (%)**	
*Cribriform plate*	4 (57%)	5 (63%)	
*Frontal sinus*	2 (29%)	1 (12%)	0.64
*Sphenoid bone/sella/suprasellar defect*	1 (14%)	2 (25%)	
*Keros classification*	**Sides (*n* = 14), *n* (%)**	**Sides (*n* = 16), *n* (%)**	
*I*	9 (64%)	3 (19%)	0.02
*II*	5 (36%)	13 (81%)	
*III*	0 (0%)	0 (0%)	
*Gera classification*	**Sides (*n* = 14), *n* (%)**	**Sides (*n* = 16), *n* (%)**	
*I*	0 (0%)	0 (0%)	1.00
*II*	11 (79%)	13 (81%)	
*III*	3 (21%)	3 (19%)	

* Fisher’s exact test, *p* < 0.05 is significant.

**Table 5 ijerph-19-13847-t005:** Comparison of outcomes between conservative and endoscopic closure.

Technique of Repair	No. of Patients, *n* (%)	Requirement of Lumbar Drain, (*n*, %)	Use of Diuretics (*n*, %)	Duration of Hospital Stay Following Surgery (Days), Mean (SD)	Follow-up Duration (Months), Mean (SD)	Recurrent Leak, (*n*, %)	*p* Value *
Conservative management	2 (13.3)	0 (0)	1 (50)	20.5 ± 0.71	3.00 ± 0.00	0 (0)	1.00
Endoscopic multilayered closure	13 (86.7)	2 (15.4)	1 (7.7)	9.07 ± 6.17	5.15 ± 4.77	1 (7.7) **	

* Fisher’s exact test, *p* < 0.05 is significant. ** Delayed leak presented one month after first repair. A subsequent second repair using a similar technique was performed successfully.

**Table 6 ijerph-19-13847-t006:** Summary of recent studies on the main outcomes of the primary endoscopic repair of CSF leaks.

Author, Year (Country)	Study Design	No of Cases (*n*)	Etiology	Site of Leak (%)	Adjunct Treatment (%)	Duration of Hospital Stays (Day)	Success Rate of Primary Endoscopic Repair (%)	Immediate Postoperative Complications (%)	Recurrent Leak (%)
Alicandri-Ciufelli, 2020 (Italy) [38]	Retrospective	29	Spontaneous	Anterior ethmoid (79), sphenoid (14), frontal (7), posterior ethmoid (7)	None	3–4	93	Recurrent meningocele without CSF leak (3.4)	7
Allensworth, 2019 (USA) [39]	Retrospective	222	Spontaneous	Cribriform (25.6), lateral sphenoid (32.4), ethmoid roof (19.8), frontal (10.6)	Acetazolamide (74.3), VP shunt (19.8), Lumbar drain (82.8)	NA	97.2	Intracranial hematoma (1) Intracranial hemorrhage (0.45) Seizure (0.45)	2.8
Bubshait, 2021 (Saudi Arabia) [40]	Retrospective	56	Spontaneous 46%, Traumatic 54%	Frontal (14), ethmoid roof (25), cribriform (39), sphenoid (21), multiple (14)	Permanent VP shunt (2)	6.5	93	NA	7
Fiore, 2021 (Italy) [41]	Retrospective	33	Iatrogenic trauma 48.5%, non-iatrogenic trauma 15.2%, spontaneous 33.3%, tumor 3%	Sphenoid (60.6), cribriform (30.3), sphenoid/ethmoid (9.1)	Lumbar drain (100)	NA	90.9	NA	9.1
He, 2020 (China) [42]	Retrospective	12	Spontaneous	Sphenoid sinus lateral recess	20% mannitol solution (100)	NA	100	None	0
Jahanshahi, 2017 (Iran) [30]	Retrospective	24	Traumatic 75%, spontaneous 25%	Frontal sinus	Acetazolamide (100)	NA	95.8	NA	4.2
Jiang, 2018 (USA) [8]	Retrospective	48	Spontaneous	Sphenoid (43.8), cribriform (33), ethmoid (17)	Lumbar drain (27), Acetazolamide (40)	NA	93.8	NA	6.2
Keshri, 2019 (India) [43]	Retrospective	43	Spontaneous	Sphenoid (6.9), cribriform (74.4), ethmoid (16.3), planum (2.3)	VP shunt/lumbar drain (53.4)	NA	95.3	Meningitis (4.7)	4.7
Kim-Orden, 2019 (USA) [44]	Retrospective	20	Spontaneous	Cribriform (44), ethmoid (32), lateral sphenoid (12), planum sphenoidale (12)	Lumbar drain (100),Acetazolamide (30)	7	92	None	8
Kreatsoulas, 2021 (USA) [31]	Retrospective	46	Spontaneous	Cribriform/ethmoid (56.6), lateral sphenoid (21.7)	permanent VP shunt (55.6), Acetazolamide (44.4)	NA	95.7	Seizures (2.2), Meningitis (4.4), Subdural hematoma (2.2)	4.3
Poma, 2021 (Italy) [45]	Retrospective	20	Traumatic	Ethmoid (65), sella (35)	NA	NA	95	None	5
Rathod, 2021 (India) [46]	Retrospective-prospective	11	Spontaneous	Sphenoid lateral recess (100)	Lumbar drain (9.1)	4	90.9	None	9.1
Sanghvi, 2020 (USA) [32]	Retrospective	33	Spontaneous	Cribriform (58), sphenoid lateral recess (30), multiple or bilateral (12)	Acetazolimide (9.1) Topiramate (3.0) VP shunt (9.1)	NA	97	None	3
Workman, 2017 (USA) [47]	Retrospective	14	Spontaneous	Cribriform (21.4), frontal (14.2), ethmoid roof (21.4), sphenoid (42.8), sphenoid/ethmoid (7.1)	Acetazolamide (92.9) Lumbar drain (100) VP shunt (21.4)	NA	85.7	NA	14.3
Xu, 2022 (China) [48]	Retrospective	15	NA	Lateral recess of sphenoid sinus (100)	20% mannitol (100)	NA	100	Temporary numbness of upper lip or cheek (33.3)	0
Zhu, 2019 (China) [49]	Retrospective	21	Spontaneous	Ethmoid roof (57), cribriform (33), sphenoid (10)	20% mannitol (100)	NA	100	Hyposmia (4.8)	0

NA—not available; VP—ventriculoperitoneal.

## Data Availability

Not applicable.

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
