# Peer review of "Management of Traumatic and Non-Traumatic Cerebrospinal Fluid Rhinorrhea—Experience from Three Southeast Asian Countries"

_ijerph, 2022, doi:10.3390/ijerph192113847_

Round 1
Reviewer 1 Report
Work review: Management of traumatic and non-traumatic cerebrospinal fluid rhinorrhea -experience from 3 Southeast Asian countries
1. Introduction
A little detailed introduction. There is no information on risk factors, frequency, or demographic factors. Explain, based on the research, why this study group was selected and not another. no clear objectives for the study
2. Materials and Methods
line78- “ The anterior skull base was evaluated according to Keros and Gera classifications”- no citation
No clear inclusion and exclusion criteria for the study. There is no information about the study group 3. Results I suggest that the data from table 1 be described in the material and methods section. The Keros and Gera classification was used for the evaluation. there is no table to compare patients type I, II and IIIline 102 “3.2 Diagnostic confirmation of CSF rhinorrhea “ this fragment of the presentation of the results is incomprehensible to me
line 127-137 in order to improve the transparency of the results, these data should be included in a table.
Line 153-162 3.4 Traumatic CSF rhinorrhea is this fragment needed at this point? Perhaps it would be worth considering listing it in the tables
line 164-170- 3.5 Non-traumatic CSF rhinorrhea as above
line 173-196 My task is to divide the patients regarding the surgical techniques and present a table in which the groups will be compared. Table 3 adds nothing new to the presentation of the results
4. Discussion In the discussion, there are few references to the research results obtained; there was also no attempt to interpret the results in the context of the latest research 5. Conclusions the conclusions must be consistent with the objectives of the work and the results presented
Author Response
Response to reviewers
#Reviewer 1
- Introduction
A little detailed introduction. There is no information on risk factors, frequency, or demographic factors. Explain, based on the research, why this study group was selected and not another. No clear objectives for the study
Response:
Thank you for the kind suggestion. We have added more information on the prevalence, risk factors and demographic of CSF rhinorrhea in the introduction.
The reason the countries of Malaysia, Singapore and Thailand are selected as the study group is due to the similarity of population demography, socioeconomic status and health care systems which could reliably outline the management practice of CSF leak from Southeast Asian countries in general. This information is added in the last paragraph of the introduction.
The objectives of this study have been rephrased to make it clear.
- Materials and Methods
line78- “The anterior skull base was evaluated according to Keros and Gera classifications”- no citation No clear inclusion and exclusion criteria for the study. There is no information about the study group
Response:
We apologize for this mistake. The citations have been added. The inclusion and exclusion criteria together with the information about the study group have been added (pages 2-3, lines 87-93).
- Results
I suggest that the data from table 1 be described in the material and methods section. The Keros and Gera classification was used for the evaluation. there is no table to compare patients type I, II and III. line 102 “3.2 Diagnostic confirmation of CSF rhinorrhea “ this fragment of the presentation of the results is incomprehensible to me. line 127-137 in order to improve the transparency of the results, these data should be included in a table. Line 153-162 3.4 Traumatic CSF rhinorrhea is this fragment needed at this point? Perhaps it would be worth considering it in the tables. line 164-170- 3.5 Non-traumatic CSF rhinorrhea as above. line 173-196 My task is to divide the patients regarding the surgical techniques and present a table in which the groups will be compared. Table 3 adds nothing new to the presentation of the results
Response:
The main data from Table 1 have been described in the material and methods section (page 3, lines 94-98). The rest of the data is presented in the results section as a Table. This we believe will make the presentation succinct and avoid redundancy.
A new table is created and labelled as Table 3 to compare Keros and Gera classifications in the assessment of skull base configuration.
Section 3.2 is relabelled as “Methods in the confirmation of site of leak” to make it clearer and the whole section has been reworded.
The data from line 127-137 were presented in table 2 and this can be compared with the newly created table 3 to aid understanding.
Both sections “3.4 Traumatic CSF rhinorrhea” and “3.5 Non-traumatic CSF rhinorrhea” have been removed as suggested and the information transferred to a new table 4.
A new table 5 has been created to compare the different treatment modalities and their outcomes as suggested.
We concur that the old table 3 is redundant and has removed it.
- Discussion
In the discussion, there are few references to the research results obtained;
there was also no attempt to interpret the results in the context of the latest research
Response:
We have expanded the discussion with explanation and comparison of our findings with other studies. Among the notable points are the comparison of the causes (traumatic versus non-traumatic; iatrogenic versus non-iatrogenic), the investigation modalities, skull base configuration according to Gera and Keros classifications and outcomes of medical and surgical treatment of CSF rhinorrhoea. Some of the added references for comparison are:
- Xie, M.; Zhou, K.; Kachra, S.; McHugh, T.; Sommer, D.D. Diagnosis and localization of cerebrospinal fluid rhinorrhea: A systematic review. Am J Rhinol Allergy 2022, 36, 397-406.
15.Prosser, J.D.; Vender, J.R.; Solares, C.A. Traumatic cerebrospinal fluid leaks. Otolaryngol Clin North Am 2011, 44, 857-873.
21.Sharma, S.D.; Kumar, G.; Bal, J.; Eweiss, A. Endoscopic repair of cerebrospinal fluid rhinorrhoea. Eur Ann Otorhinolaryngol Head Neck Dis 2016, 133, 187-190.
22.Psaltis, A.J.; Schlosser, R.J.; Banks, C.A.; Yawn, J.; Soler, Z.M. A systematic review of the endoscopic repair of cerebrospinal fluid leaks. Otolaryngol Head Neck Surg 2012, 147, 196-203
- Conclusions
the conclusions must be consistent with the objectives of the work and the results presented
Response:
The conclusions have been rephrased to be consistent with the objectives and findings of the study.

Reviewer 2 Report
Minor comments:
1) line 33: Cand, is this a typo?
2) line 82: bracket was opened but not closed, please check.
Major comments:
The major limitation of this article is the number of subjects as highlighted by authors this can be improved by adding retrospective literatures from other publication in this field and their findings in the discussion section.
Author Response
#Reviewer 2
- Minor comments:
1) line 33: Cand, is this a typo?
2) line 82: bracket was opened but not closed, please check.
Response:
We are sorry for the typo errors. Both have been corrected.
- Major comments:
The major limitation of this article is the number of subjects as highlighted by authors this can be improved by adding retrospective literatures from other publication in this field and their findings in the discussion section.
Response:
We concur with this thoughtful comment. We have added comparison of our findings with others. Among the notable comparison is two studies [2,21] involving 1622 and 2000 patients respectively with CSF rhinorrhoea. This will give a great overall coverage of the subject and better perspective of the relevant issues.
- Xie, M.; Zhou, K.; Kachra, S.; McHugh, T.; Sommer, D.D. Diagnosis and localization of cerebrospinal fluid rhinorrhea: A systematic review. Am J Rhinol Allergy 2022, 36, 397-406.
- Sharma, S.D.; Kumar, G.; Bal, J.; Eweiss, A. Endoscopic repair of cerebrospinal fluid rhinorrhoea. Eur Ann Otorhinolaryngol Head Neck Dis 2016, 133, 187-190.

Round 2
Reviewer 1 Report
Thank you very much for making changes in the work, but in my opinion it still does not fully meet the criteria of research work The statistical analysis is a big disadvantage of the work. Percentage distribution is not a method that fully illustrates the results. Moreover, in the discussion there are too few references to the current available research illustrating the discussed topicAuthor Response
Thank you very much for making changes in the work, but in my opinion it still does not fully meet the criteria of research work. The statistical analysis is a big disadvantage of the work. Percentage distribution is not a method that fully illustrates the results.
Response: Thank you for your valuable comment and suggestion. We have added statistical analysis to Table 4 and 5. Relevant details have been added in the methodology and results section.
Moreover, in the discussion there are too few references to the current available research illustrating the discussed topic.
Response: We searched via PUBMED for relevant studies within the past 5 years for comparison. We created a new table (Table 6) to summarize the findings which will help readers to appreciate the key comparison in the overall studies. A total of 13 new citations have been added in the discussion.
If there are any studies that we have missed, please let us know and we will redress it. We appreciate your time and effort in improving the quality of our manuscript. Thank you.
